# Influence of Scaffold Microarchitecture on Angiogenesis and Regulation of Cell Differentiation during the Early Phase of Bone Healing: A Transcriptomics and Histological Analysis

**DOI:** 10.3390/ijms24066000

**Published:** 2023-03-22

**Authors:** Julien Guerrero, Ekaterina Maevskaia, Chafik Ghayor, Indranil Bhattacharya, Franz E. Weber

**Affiliations:** 1Oral Biotechnology and Bioengineering, Center for Dental Medicine, University of Zurich, 8032 Zurich, Switzerland; 2Center for Applied Biotechnology and Molecular Medicine, University of Zurich, 8057 Zurich, Switzerland

**Keywords:** tricalcium phosphate, osteoconduction, microarchitecture, bone substitute, additive manufacturing, 3D printing, angiogenesis, transcriptomic analysis, RNAseq

## Abstract

The early phase of bone healing is a complex and poorly understood process. With additive manufacturing, we can generate a specific and customizable library of bone substitutes to explore this phase. In this study, we produced tricalcium phosphate-based scaffolds with microarchitectures composed of filaments of 0.50 mm in diameter, named Fil050G, and 1.25 mm named Fil125G, respectively. The implants were removed after only 10 days in vivo followed by RNA sequencing (RNAseq) and histological analysis. RNAseq results revealed upregulation of adaptive immune response, regulation of cell adhesion, and cell migration-related genes in both of our two constructs. However, significant overexpression of genes linked to angiogenesis, regulation of cell differentiation, ossification, and bone development was observed solely in Fil050G scaffolds. Moreover, quantitative immunohistochemistry of structures positive for laminin revealed a significantly higher number of blood vessels in Fil050G samples. Furthermore, µCT detected a higher amount of mineralized tissue in Fil050G samples suggesting a superior osteoconductive potential. Hence, different filament diameters and distances in bone substitutes significantly influence angiogenesis and regulation of cell differentiation involved in the early phase of bone regeneration, which precedes osteoconductivity and bony bridging seen in later phases and as consequence, impacts the overall clinical outcome.

## 1. Introduction

Bone healing is a complex physiological process closely regulated by a large variety of different growth factors, transcription factors, hormones, and cytokines [1]. It takes place mainly in four consecutive and overlapping phases, which include hematoma formation with inflammatory response, fibrocartilaginous callus formation with the development of new blood vessels, bony callus formation, and finally bone remodeling [2]. In detail, it starts with an inflammatory reaction [3], in which recruited immune cells release inflammatory cytokines [4], thus initiating the healing process. Next, revascularization (blood vessel recruitment and infiltration) together with homing, proliferation, and differentiation of mesenchymal stromal cells (MSCs) are key events to initiate a successful regenerative and healing process [5]. Later, the development of a soft callus gives some stability back to the injured load-bearing structure. Then, fibrous tissue develops into fibrocartilage and subsequently into hyaline cartilage before mineralization occurs. After the formation of the hard callus, a remodeling phase begins, which can last for months or even years, adapting the bone inner structure to the mechanical strain it encounters during weight-bearing [2].

In most cases, those finely tuned sequences reach complete bone healing; however, non-union or even delayed bone tissue healing can be observed in up to 10% of patients in clinics [6], posing a major clinical challenge. A deeper understanding of the causes of unsuccessful healing is essential in current treatment and may even lead to new treatment strategies. More precisely, a better and deeper understanding of the early phase of bone healing in the context of the microarchitecture of bone substitutes, with the help of transcriptomic analysis, could increase our understanding of the interplay between scaffold microarchitecture [7] and angiogenesis [8] in bone regeneration [9]. Moreover, such results might even be transferred to other scaffold-based tissue regeneration concepts since the majority are dependent on angiogenesis.

The process of bone regeneration has been extensively investigated in fracture studies, where both intramembranous and endochondral ossification are usually observed [10]. In this context, a critical-size defect (CSD) is defined as being too large to heal over the normal life span of the organism, resulting in a non-union [11]. Critical-size but also smaller defects can be used as experimental models to test different procedures or biomaterials for bone regeneration, such as guided bone regeneration (GBR) [12], bone grafts, or implantation of bone substitutes for tissue engineering and bone regeneration procedures. In this context, defects in long bones have to be stabilized. To avoid the use of fixation systems and their possible interference with bone healing, calvarial defects are frequently applied as an in vivo model, due to easy access to the skull, the possibility of the creation of multiple, standardized defects in a single animal, and the lack of mechanical stress [13,14]. The disadvantage of a calvarial bone defect model is that bone regeneration mainly follows the intramembranous route under no-load conditions.

In the past years, gene expression analyses have been performed in several studies on bone fractures [15,16,17] and bone loading [18,19,20]. Those studies of genome-wide expression in bone healing have demonstrated the complexity of osseous regeneration and have started to elucidate some of the key signaling pathways involved [16,17,21,22]. We can notably cite the adaptive immune response as an important phase in early bone tissue healing, since it is responsible for the activation of various growth factors and cytokines which attract the osteogenic cells into the wound and regulate the healing process [23,24]. Angiogenesis is a key signaling pathway involved in bone healing with genes such as *ANGPTL4*, a member of the angiopoietin-like gene family, which is upregulated at this early healing stage. Another upregulated gene linked to angiogenesis is T-cadherin (also known as *CDH13*), which is already documented as a mediator of intracellular signaling in vascular cells, being expressed by endothelial cells and being critical within adipose tissue to regulate osteogenesis [25]. However, to the best of our knowledge, no study has described gene expression as early as 10 days in a model of intramembranous ossification where the microarchitecture in terms of filament thickness and distance is the only variant.

Here we present a study in which scaffolds were designed and made with additive manufacturing, allowing their customization from their macroarchitecture down to their microarchitecture levels (Figure 1). The macroarchitecture of a bone defect is defined as the defect volume [26] when the macroarchitecture of a bone substitute should perfectly fit the defect to become a personalized bone substitute [27]. The microarchitecture is defined as the distribution of the material in the macroarchitecture. For autologous bone substitutes it describes the distribution of cortical bone and trabeculae of cancellous bone [28]. The nanoarchitecture is the third level of architecture and for bone substitutes, it characterizes their microporosity, grain size, and surface roughness [29,30]. For autologous bone substitutes, such parameters are not applicable and the size of collagen molecules and mineral crystals present in the mineralized collagen fibril of the bone tissue [31] is of a different magnitude.

In the early days of the use of synthetic bone substitutes the ideal porosity was set at 75 to 85% and pore size between 0.3 and 0.5 mm in diameter to reflect cancellous bone porosity, which is still the gold standard for bone substitutes in the clinic [32]. However, by application of additive manufacturing, pore-based scaffold microarchitectures were recently evaluated and 0.8 to 1.2 mm is suggested as the ideal pore diameter for osteoconduction [7]. Osteoconduction is defined as the ingrowth of sprouting capillaries, perivascular tissue, and osteoprogenitor cells from a bony bed into the 3D structure of a porous implant which serves as a guiding cue to bony bridge the defect [26,33,34]. Key players during bone regeneration are pericytes, mesenchymal stromal cells, and endothelial cells, which guide osteoconduction by biophysical and biochemical cues triggered by the microenvironment [35]. Cell guiding within the bone substitute is a process that involves cell adhesion, polarization, and movement in a predefined direction [36,37,38].

It has been demonstrated that several parameters of 3D printed scaffolds could influence cell behavior, such as directionality, as reported in fibers from electrospun samples within the range of 100 nm to 1000 nm shown to guide cell migration [39]. Another parameter, transparency (i.e., the relative free area in the projection of the scaffold in the different spatial directions), was shown to affect new bone formation in an implanted scaffold by facilitating a straight ingrowth [26]. However, to our knowledge, there are no studies where the modulation of scaffold microarchitecture by the filament dimension and distance was investigated down to the molecular level during the early phase of bone healing.

In this study, we used two tricalcium phosphate (TCP)-based bone substitutes with microarchitectures formed by a layered arrangement of 0.50 mm or 1.25 mm filaments, named Fil050G and Fil125G, respectively. In a recent study, those two scaffolds were established with marked differences in their osteoconductivity resulting in high (Fil050G) or low (Fil125G) osteoconductive scaffolds, although the porosity, microporosity, transparency and degree of directionality of the filaments was identical [40]. Since the initial study was based solely on 4-week in vivo data, we now aimed towards an early phase of 10 days after implantation, to investigate the fundamental difference leading to high or low bony bridging or osteoconductivity at 28 days after implantation. To that end, we performed RNAseq and Gene Ontology enrichment analysis of Fil050G and Fil125G samples harvested 10 days after implantation, to evaluate the effect of filament dimension and distance on (i) biological processes, (ii) molecular functions, and (iii) cellular components at an early phase of bone regeneration. Additionally, histological and immunohistological analyses were performed to better characterize cells and structures involved in the early process with emphasis on vascularization.

## 2. Results

### 2.1. Gene Ontology

According to the Gene Ontology (GO) system, gene names resulting from the RNAseq data can be clustered into three different categories, namely “biological process”, “molecular function”, and “cellular component” [41]. In this study, the *Homo sapiens*, *Rattus norvegicus*, and *Mus musculus* databases were used and investigated as a triple screening based on our samples extracted from Oryctolagus cuniculus.

The main biological processes differentially expressed at day 10 in both of our bone substitutes are listed in Table 1. We identified upregulation of the genes linked to the adaptive immune response, regulation of cell adhesion, and cell migration together with other processes, such as anatomical structure development and muscle system (in the *Homo sapiens*, *Rattus norvegicus*, and *Mus musculus* databases). Furthermore, cell motility, regulation of cell signaling, and regulation of cell communication were also detected in two out of three databases (*Rattus norvegicus* and *Mus musculus*). Additionally, some processes, mainly related to smell perception, response to chemical stimulus, sensory perception, and metabolic process were only represented in the *Rattus norvegicus* database.

Moreover, looking directly at the comparison of the “biological process” between our high osteoconductive Fil050G scaffold and low osteoconductive Fil125G scaffold in this early phase of bone defect healing, genes overexpressed and linked to angiogenesis and regulation of cell differentiation were observed (Table 2) in Fil050G but not in Fil125G samples. Additionally, developmental processes were overexpressed on day 10 only in the Fil050G-related scaffolds, including muscle tissue development, mesoderm formation, mesoderm morphogenesis, and skeletal muscle tissue development, as shown in Table 2.

When clustering GO terms according to the “molecular function”, only terms related to Fil050G were observed (Table 3). First, both phospholipase and lipase activity were detected and are overexpressed in all three databases. Furthermore, we also notably detected calcium-dependent phospholipase A2 activity, phospholipase A2 activity, carboxylic ester hydrolase activity, nuclear glucocorticoid receptor binding, protein binding, metal ion binding, cation binding, and olfactory receptor activity in two out of the three databases (*Rattus norvegicus* and *Mus musculus*). Additionally, actin binding and ion binding together with odorant binding were detected to be upregulated in one out of three databases (*Rattus norvegicus* and *Mus musculus*, respectively).

The clustering of GO terms according to the “cellular components”, produced matches with Fil050G-related samples only (Table 4). At 10 days we noted the extracellular matrix (ECM)-related terms with the presence of collagen-containing ECM and external encapsulating structure in the three consulted databases (*Homo sapiens*, *Rattus norvegicus*, and *Mus musculus*). Moreover, cellular activity discriminated Fil050G from Fil125G samples, with the overexpression of plasma membrane-bounded cell projection, cell projection, and cellular anatomical entity in two of three databases (*Rattus norvegicus* and *Mus musculus*). Furthermore, we detected sarcolemma, extracellular region, ribonucleoprotein complex, and caveola for the *Rattus norvegicus* database and cell periphery component for the *Mus musculus* database.

### 2.2. Pathway Analysis in Fil050G (Highly Osteoconductive) Scaffolds in the Early Phase of Bone Healing

We investigated “biological processes” with genes only overexpressed in the Fil050G samples compared to Fil125G after 10 days of implantation. We targeted our analysis on the angiogenesis and regulation of cell differentiation pathways as key elements involved within the osteoconductive process of our scaffold.

#### 2.2.1. Angiogenesis

The GO enrichment analysis linked to the transcriptomic analysis revealed gene upregulation linked to angiogenesis with a variety of fold increases (Figure 2). In all the genes detected (Figure 2A), the most upregulated were Proepiregulin (*EREG*) with a 10.53-fold increase, Nuclear receptor subfamily 4 group A member 1 (*NR4A1*) with a 4.53-fold increase, Fibroblast growth factor 18 (*FGF18*) with a 3.04-fold increase, C-X-C chemokine receptor type 3 (*CXCR3*) with a 2.95-fold increase, and Fibroblast growth factor 10 (*FGF10*) with a 2.63-fold increase. Interestingly, Angiopoietin-related protein 4 (*ANGPTL4*) was also overexpressed in Fil050G samples compared to Fil125G with a 2.61-fold increase (Figure 2B).

#### 2.2.2. Regulation of Cell Differentiation

In genes upregulated and linked to the regulation of cell differentiation (Figure 3A) several of them were associated with mesenchymal stromal cell differentiation. We can cite 5-hydroxytryptamine receptor 2A (*HTR2A*) with a 5.20-fold increase, Myocilin (*MYOC*) with a 3.09-fold increase, ABI Family Member 3 Binding Protein (*ABI3BP*) with a 3.87-fold increase, Histone acetyltransferase KAT2A (*KAT2A*) with a 2.30-fold increase, Chemerin Chemokine-Like Receptor 1 (*CMKLR1*) with a 2.40-fold increase, and Tuberin (*TSC2*) with a 2.11-fold increase (Figure 3B).

#### 2.2.3. Bone Formation

At day 10, terms highlighted by GO enrichment analysis and linked to bone or mineralization were not significantly overexpressed in any of our sample conditions (i.e., Fil050G and Fil125G). However, many genes being detected by GO enrichment analysis in the “angiogenesis” and “regulation of cell differentiation” biological processes and overrepresented in the Fil050G-related samples could be linked to bone development and ossification biological processes (Table 5). We can notably cite *MYOC*, *FGF18*, and Parathyroid hormone (*PTH*) with a 4.52-fold increase as the most upregulated between the Fil050G and Fil125G samples.

### 2.3. Analysis of Relative Gene Expression by qPCR in the Early Phase of Bone Healing

To investigate results obtained with RNAseq in combination with the GO enrichment analysis, we performed qPCR analysis on genes presenting the highest fold increase between Fil050G and Fil125G samples in a larger sample population (Figure 4). The two genes that were the most upregulated on the RNAseq analysis in Fil125G samples compared to Fil050G, namely NADH-ubiquinone oxidoreductase chain 4L (*MT-ND4L*) and Translocator Protein 2 (*TSPO2*), were used as the control (Figure 4). In addition, we performed qPCR analysis on genes related to the osteogenic pathway (Figure 5). All qPCR analyses were based on rabbit (*Oryctolagus cuniculus*) primers (Table 6).

#### 2.3.1. Cell Differentiation/Mesenchymal Stromal Cells Differentiation

First of all, we investigated genes involved in the regulation of cell differentiation linked with mesenchymal stromal cells. As observed with the RNAseq analysis, we detected significant overexpression of *ABI3BP* (Figure 4A), *CMKLR1* (Figure 4B), *KAT2A* (Figure 4C), *MYOC* (Figure 4D) and *TSC2* (Figure 4E) in our 12-sample comparison (six Fil050G- and six Fil125G-related samples).

#### 2.3.2. Angiogenesis

For genes linked to angiogenesis, we investigated *FGF10* (Figure 4F) and *ANGPTL4* (Figure 4G). As shown by the RNAseq analysis, both of them were significantly upregulated after 10 days in the comparison of our six Fil050G-related samples with the six Fil125G-related samples.

#### 2.3.3. Highly Expressed in Fil125G

As a control and to confirm our precedent finding from RNAseq with rabbit primers, we also investigated genes highly overexpressed in Fil125G-related samples. We studied *MT-ND4L* (Figure 4H) and *TSPO2* (Figure 4I). Both of them were significantly upregulated in the Fil125G-related samples compared to Fil050G-related samples after 10 days of implantation.

#### 2.3.4. Bone Formation

With the previous indications, RT-q-PCR was run on osteogenic-related genes to study a higher number of replicates for differences between Fil050G and Fil125G scaffolds (Figure 5). We did not observe any differences for alkaline phosphatase (*ALPL*) (Figure 5A), runt-related transcription factor 2 (*RUNX2*) (Figure 5C), and caveolin 1 (*CAV*1) (Figure 5F). However, for the collagen type I (*COL1A1*) (Figure 5B), osteopontin (*OPN*) (Figure 5D), and osterix (*SP7*) (Figure 5E) a significantly higher expression in the highly osteoconductive Fil050G scaffold in comparison with the Fil125G scaffold was observed.

### 2.4. Mineralized Tissue after 10 Days of Implantation

To investigate the osteoconduction within our two types of scaffolds in the early phase of bone healing, we performed a quantified µCT analysis (Figure 6). In the pictures with µCT, we were able to identify (red arrow) infiltration of mineralized tissue in several areas of Fil050G-related samples (Figure 6A–D). However, in the Fil125G-related samples, mineralized tissue was only observed in the surrounding of the defects with no infiltration/migration within the defect area (Figure 6E,F). After quantification, we observed that the percentage of mineralized tissue (Figure 6I) and the ratio of mineralized tissue per volume of interest (Figure 6J) were both significantly higher in Fil050G-related samples compared to Fil125G-related samples.

### 2.5. Blood Vessel Analysis after 4 Weeks of Implantation

Knowing the overrepresentation of angiogenesis-related genes in “biological process” terms after 10 days based on RNAseq analysis, we decided to investigate the number of blood vessels on histological sections from our two types of samples after 10 days (paraffin sections) and 4 weeks (MMA sections) of implantation.

#### Microvasculature and Capillaries

To explore the impact of microarchitecture on blood vessel infiltration within our two types of scaffolds, we performed histological staining on paraffin sections with an antibody targeting Laminin epitope (Figure 7). After 10 days, we observed in Masson’s Trichrome stained sections that defects were filled with the scaffold and surrounded by bone tissue stained in deep green (Figure 7A,E, for Fil050G and Fil125G, respectively) with infiltration of tissue already up to the center of each scaffold.

Furthermore, we performed Laminin immunohistological staining to label the blood vessel structure on our section (Figure 7B–D,F–H, for Fil050G and Fil125G, respectively). We divided the histological section into three different areas identified as follows: outside of the defect (Figure 7B,F), edges of the defect/scaffold (Figure 7C,G), and the center of the defect/scaffold (Figure 7D,H). For both conditions (i.e., Fil050G and Fil125G) we detected several structures positive for Laminin labeling in the area located outside of the defect/scaffold (Figure 7B,F). The same process was performed for the Laminin structure on the edges of the defect/scaffold for both conditions. However, on day 10, the presence of Laminin-positive structures was only detected in Fil050G-related samples (Figure 7D) but not in Fil125G-related samples (Figure 7H).

To confirm our previous observation, we counted the number of Laminin-positive structures on the paraffin histological section after 10 days of implantation (Figure 8A–C) and the number of blood vessels within scaffolds on the MMA section after 4 weeks of implantation (Figure 8D–F). The number of structures positive for Laminin was not significantly different for the area located outside of the area of interest (Figure 8A). Nonetheless, for Laminin-positive structures in the edges of the AOI (Figure 8B) and the center of the AOI (Figure 8C), we found a significantly higher number in Fil050G samples compared to Fil125G-related samples. Additionally, the counting performed after 4 weeks on the MMA section showed a significantly higher number of blood vessels in the Fil050G samples compared to Fil125G-related samples under brightfield light with identification of circular structures with a lumen (Figure 8D), fluorescence light with erythrocyte autofluorescence (Figure 8E), and the mean of both (Figure 8F).

## 3. Discussion

The calvarial defect is a well-described and used model for studying a large variety of microarchitectures of bone substitutes produced with additive manufacturing [40]. Derived from earlier studies, we chose two filament-based microarchitectures, categorized as high (Fil050G) or low (Fil125G) osteoconductive [40] based on the percentage of the bony regenerated area of the defect at 4 weeks of healing. Variables between both microarchitectures are the filament thickness and the distance between the filaments of 0.50 mm or 1.25 mm, respectively. Other characteristics such as material, amount of material, porosity, microporosity, degree of directionality of the filaments, and degree of transparency were identical. The present study aimed to unravel the influence of high and low osteoconductive microarchitectures in terms of filament thickness and distance on the gene expression profiles in the early phase of bone healing and osteoconduction.

The 10-day time point was chosen since it covers the stage when the initial inflammation is gradually overlapped by a progressive infiltration by endothelial and stromal cells into the defect area [17,42,43,44]. Several “biological processes” terms such as adaptive immune response, regulation of cell adhesion, and cell migration were clearly overrepresented at 10 days irrespective of the scaffold’s microarchitecture or osteoconductivity level (Table 1). Those pathways were already identified as part of the early phase of bone healing in GBR-treated calvarial critical-size defects in rats [22] and several in vivo fracture healing studies [45,46,47]. Moreover, they were already documented as beneficial in the regenerative process [48,49,50] and confirmed the suitability and biocompatibility of both scaffold types for calvarial bone tissue regeneration.

In the current investigation, the RNAseq analysis provided information on the expression of more than 20,000 genes. Regarding the “biological processes” differentially expressed between our two conditions, we found angiogenesis to be overrepresented solely in the Fil050G samples (Table 2). The upregulated genes representing angiogenesis (Figure 2) were already described as being essential and critical for a successful healing process. Briefly, *FGF10* is involved in the formation of blood vessels, affects both vasculogenesis and angiogenesis in vitro and in vivo [51], and is upregulated in Fil050G 2.63-fold compared to Fil125G samples. The most upregulated gene from the “biological process” of angiogenesis identified here is Proepiregulin (*EREG*). It is known as a positive angiogenic regulator that can be induced in hypoxic conditions [52], which was potentially the case at the center of our constructs before blood vessel density reached a sufficient concentration for O_2_ perfusion. Since scaffold porosity and microporosity are the same for both scaffolds, a lower O_2_-concentration in Fil050G samples appears unlikely and additional factors might have induced EREG expression. Moreover, *NR4A1* and *FGF18*, also upregulated in Fil050G samples, have links to angiogenesis/osteogenesis-related factors [53,54,55]. Furthermore, *ACVRL1*, C-type lectin domain family 14 member A (*CLEC14A*) [56], and Nitric oxide synthase endothelial (*NOS3*) [57] possess angiogenic/vasculogenic potential within the bone healing process [58].

As angiogenesis is a critical parameter in bone healing [59,60] and was overrepresented on day 10, we next looked at the possible result of the elevated expression of *ANGPTL4*, *FGF10*, and *CDH13* (Figure 3 and Figure 4) by quantifying blood vessel formation. Blood vessel formation is a well-recognized requirement for regeneration, where angiogenic factors may interact with stromal bone cells to promote osseous formation [61]. The final manifestation of angiogenesis as the number of blood vessels correlated well with the osteoconductivity of the microarchitecture (Figure 7 and Figure 8). Therefore, the microarchitecture of scaffolds orchestrates late-phase osteoconductivity and bony healing of the defect [40] via early-phase angiogenesis. It appears that guided growth of blood vessels through a scaffold is facilitated by 50% of 0.50 mm filaments in line with defect bridging direction and hampered by 50% of 1.25 mm filaments in the same direction. Since the area devoid of material, indicated as porosity and microporosity and transparency, is identical in both constructs, vessel growth is directly related to filament orientation, size, and distance. This marked difference between 0.50 mm and 1.25 mm is particularly surprising since for vascularization itself, pores between 0.04 mm and 0.07 mm would be sufficiently wide [62]. One could speculate that with increased distance between the filaments, the guiding of blood vessel formation by filaments is insufficient. Angiogenesis and osteogenesis are highly interweaved. In this context, *ANGPTL4* was already shown to be expressed as early as 3 days into defect healing, was detected in areas of condensing MSCs, mineralizing osteoblasts, and new bone formation [63], and is described as a potential key regulator of osteoblastic differentiation and angiogenesis [64,65]. In our study, *ANGPTL4* was upregulated in Fil050G over Fil125G by the factor 2.61-fold increase (Figure 4) and links the microarchitecture-specific increase in angiogenesis with osteoblastic cell differentiation.

Concerning differentiation of MSCs, it was not possible to detect the overrepresentation of enough genes to reveal through Gene Ontology enrichment analysis the term “ossification”. However, after qPCR analysis, we were able to observe a significant upregulation of *ALPL*, *COL1A1*, *OPN*, and *SP7* (Figure 5) confirming the more advanced state of osteoblastic differentiation of the resident cells in the Fil050G samples compared to Fil125G-related samples. This was further supported by µCT measurements, since a significantly higher amount of mineralized tissue was found in Fil050G samples.

In our study, *HTR2A,* a serotonin receptor known to be present in osteoblast [66] and involved in their proliferation [67], showed the highest fold increase in the Fil050G- compared to Fil125G-related samples (Figure 3). This correlated well with the higher amount of mineralized tissue we detected in Fil050G samples by µCT (Figure 6K,L). Furthermore, *PTH*—known to induce differentiation of mesenchymal stromal cells within the osteogenic lineage and to stimulate bone remodeling [68]—was overexpressed in the osteoconductive Fil050G scaffold (Figure 3). Additionally, genes investigated in a larger sample population by qPCR were shown to be involved in new bone formation and/or differentiation of MSCs. Briefly, *ABI3BP* is known as a regulator of MSC biology [69], *CMKLR1* as being involved in osteoblastogenesis of MSCs [70], *KAT2A* as being involved in craniofacial cartilage and bone growth [71], and *MYOC* is known for its role in the osteogenesis of MSCs [72]. Altogether, variations of microarchitectures of bone substitutes in terms of an increase in filament thickness and a distance from 0.50 to 1.25 mm at otherwise high similarity are sufficient to orchestrate defect healing and osteoconductivity, not only by enhanced angiogenesis but also by inducing cell differentiation towards the osteogenic lineage.

Concerning “molecular function” terms, phospholipase and lipase activity were found to be overrepresented in the GO enrichment analysis in Fil050G- but not in Fil125G-related samples. Since phospholipase and lipase activity are involved in bone biology and fracture healing [73,74], it further supports the significantly higher osteoconductive potential of the Fil050G scaffold in the early phase of bone healing. Moreover, the overrepresented terms of “cellular components”, such as collagen-containing ECM, ECM, and external encapsulating structure found for the osteoconductive Fil050G samples, are linked to the new formation of bone tissue [75] and correlate well with our µCT results (Figure 6K,L).

## 4. Materials and Methods

### 4.1. Scaffold Production

The scaffolds were composed of unit cells formed by cubes of 1.00 or 2.50 mm in length to build filament-based scaffolds mimicking filaments of 0.50 or 1.25 mm in a square, as already documented [40]. The TCP scaffolds were produced with TCP slurry LithaBone™ TCP 300 (Lithoz, Vienna, Austria) as previously described [30]. In brief, the CeraFab 7500 system (Lithoz, Vienna, Austria) was used to form a green body from the slurry by exposing each layer (25 µm) to a blue LED light at a resolution of 50 µm in the x/y-plane. After the layer-by-layer building of the green body, it was removed from the printer’s building platform with a razor blade cleaned with LithaSol 20™ (Lithoz, Vienna, Austria), and dried with pressurized air. Next, in a heat-driven process, the polymeric binder was decomposed, and the remaining ceramic particles were sintered to increase the density with a dwell time of 3 h at 1100 °C. The resulting sintered TCP scaffolds were then transferred onto a sterile bench, packed for incorporation into the operation workflow, and used as implants for rabbit non-critical calvaria defects without further sterilization.

### 4.2. Surgical Procedure

Eighteen full-grown (6–8 months old) female New Zealand white rabbits were used in this study [29]. Animal weights ranged from 3.5 to 4.5 kg, and all animals were fed following a standard laboratory diet. The procedures (Figure 9) were evaluated and accepted by the local authorities (65/2018; 90/2021) and are in line with the EU Directive 2010/63/EU for animal experiments. In brief, before surgery, animals were anesthetized by an injection of ketamine (65 mg/kg) and xylazine (4 mg/kg). The anesthesia was maintained during the operation with a mix of isoflurane and O_2_. Next, the skin on top of the cranium was disinfected with Braunol and an incision was made from the nasal bone to the mid-sagittal crest (Figure 9A). After this, the soft tissue was deflected and fixed, and the periosteum was removed (Figure 9B). By the use of a 6.00 mm trephine bur, four defects were marked in the rabbit’s calvaria. Inside this mark, all defects were completed with rose burrs of 5.00 mm in diameter, followed by a burr with a 1.00 mm diameter to preserve the dura (Figure 9C). To remove any bone debris, defects were flushed with saline solution, then implants were applied by gentle press-fitting. Finally, the closing of the wound was performed with sutures (Figure 9D). Each animal received both implants (i.e., Fil050G and Fil125G-based scaffold types). Treatment modalities were assigned to random positions in the first animal, and thereafter, cyclically permuted clockwise. Treatment conditions were labeled Fil050G and Fil125G.

### 4.3. RNA Extraction and Purification

At 10 days post-implantation, rabbits received general anesthesia and were sacrificed by an overdose of pentobarbital. The cranial section containing all four craniotomy sites was removed and placed in a solution of RNAlater™ (#AM7021, Invitrogen, Waltham, MA, USA) prior to further procedures. Craniotomies were rapidly dissected to remove soft tissue, then implants were removed, collected, and separately stored in 2 mL sterile tubes prefilled with 500 µL of QIAzol Lysis Reagent (#79306, Qiagen, Hilden, Germany), and twice 2.4 mm sterile metal beads (VWR International, Radnor, PA, USA). The implanted samples were either stored at −80 °C for RNA extraction and purification or directly used.

Our method of RNA extraction was adapted and modified based on already described protocols [76,77]. The full process of RNA isolation from implanted scaffolds was performed using the RNeasy Mini Kit (#74004, Qiagen, Hilden, Germany). Briefly, once the samples were removed from −80 °C, 500 µL of cold QIAzol Lysis Reagent was immediately added to each sample. Then, the samples were crushed and homogenized using a tissue lyser (TissueLyser II, Qiagen, Hilden, Germany) for 3 min at 30 Hz. A total of 200 μL of chloroform was added to the samples and vortexed for 15 s. Next, the samples were centrifuged for 15 min at 4 °C (10,000× *g*). The aqueous phase, approximately 700 μL, was removed and added to 700 μL of 100% ethanol to precipitate nucleic acids. RNA purification was then performed using an RNeasy Mini Kit according to the manufacturer’s instructions, including DNase digestion with an RNase-free DNase kit (#79254, Qiagen, Hilden, Germany). At the end of the process, a final volume of 50 μL was eluted with RNAse-free water (#129112, Qiagen, Hilden, Germany).

RNA purity and quantity were determined using a spectrophotometer (Nanodrop 2000 Spectrophotometer, Thermo Fisher Scientific, Waltham, MA, USA). In addition, RNA integrity number (RIN) and quality parameters were assessed as quality control by the company performing the transcriptomic analysis (Qiagen, Hilden, Germany).

### 4.4. RNA Sequencing

The library preparation was performed using the QIAseq Stranded mRNA Select Kit with FastSelect rRNA depletion (Qiagen, Hilden, Germany). The mRNA was enriched from a 1000 ng starting material. The RNA was fragmented using enzymatic fragmentation. After first and second-strand synthesis, the cDNA was end-repaired and 3′ adenylated. Sequencing adapters were ligated to the overhangs. Adapted molecules were enriched using 11 cycles of PCR and purified by a bead-based cleanup. Library preparation was quality controlled using capillary electrophoresis (Agilent DNA 1000 Chip, Santa Clara, CA, USA). High-quality libraries were pooled based on equimolar concentrations. The library pool(s) were quantified using qPCR and the optimal concentration of the library pool was used to generate the clusters on the surface of a flowcell before sequencing on a NextSeq (Illumina Inc, San Diego, CA, USA) instrument (1 × 75, 2 × 8) according to the manufacturer’s instructions (Illumina Inc, San Diego, CA, USA).

### 4.5. Gene Ontology (GO) Enrichment Analysis

Gene Ontology (GO) enrichment analysis was performed online (http://geneontology.org/, (accessed on 15 January 2023)) and was used for interpreting RNAseq data and generating hypotheses about the underlying biological phenomena of our experiments [78,79]. The GO analysis included biological processes, molecular function, and cell composition (*p* < 0.05, statistically significant). The PANTHER (Protein Analysis Through Evolutionary Relationships) classification system version 17.0 was used to study functional relationships between protein-coding genes [80].

### 4.6. Quantitative Real-Time Polymerase Chain Reaction

Reverse transcription of 1 µg RNA was performed using iScript™ Reverse Transcription Supermix (#1708840, BioRad, Hercules, CA, USA) according to the manufacturer’s recommendations. The polymerase chain reaction was realized with each sample in duplicates using a Bio-Rad CFX96 Real-Time System and SYBR^®^ Green Supermix (#1708880, BioRad, Hercules, CA, USA) using specific primers. Gene expression was normalized to two reference genes: ribosomal protein s18 (also known as *18S*) and glyceraldehyde 3-phosphate dehydrogenase (also known as *GAPDH*) using the comparative 2^−∆∆CT^ method as already described [81]. The primers used in this study (Table 6) were commercially available (BioRad, Hercules, CA, USA).

### 4.7. Histomorphometry by µCT

After Methylmethacrylate (MMA) embedding, µCT scanning of the samples was performed using SkyScan 1272 (Bruker, Kontich, Belgium) with the following parameters: voltage of 90 kV, current of 111 µA, Al 0.5 + Cu 0.038 filter, pixel size of 10 µm, 360° scan with the rotation step of 0.3°. After the reconstruction made with NRecon, the images were visualized using DataViewer. The amount of mineralized tissue was quantified with the use of CTAn with the same thresholding for all scans and was further normalized to the height of the scaffold or to the volume of interest (VOI), which was defined as the inner part of the scaffold. All software was provided by Bruker (Bruker, Kontich, Belgium).

### 4.8. Assessment of In Vivo Vascularization by Immunohistofluorescence

The in vivo presence of blood vessels and vascularization was assayed as follows. After 10 days, implanted scaffolds were harvested with the surrounding calvarial tissue, fixed overnight in 10% formalin, completely decalcified with EDTA-based solution at 37 °C, and paraffin-embedded as previously described [25]. Sections (10 μm) were sliced along the horizontal plane of the scaffold. Then, rehydration was performed, and samples were stained with Laminin antibody specific for rabbit epitope at 1:100 (#33-5300, Thermo Fischer Scientific, Waltham, MA, USA). Fluorescent-conjugated goat anti-mouse Alexa 488 (#A-11001, Thermo Fischer Scientific, Waltham, MA, USA) was used at 1:250 as a secondary antibody. Counterstaining was performed with a DAPI solution at 1:1000 (#62248, Thermo Fischer Scientific, Waltham, MA, USA). Stained sections were examined under a fluorescent microscope (ZOE fluorescent cell imager, BioRad, Hercules, CA, USA) to acquire pictures and quantify the Laminin-positive structure. Fluorescent field images of sections of each construct were acquired from outside of the calvarial defect, at the edges of the calvarial defect, and in the center of the calvarial defect. These three regions were used to count Laminin-positive structures (i.e., blood vessels) in Fil050G and Fil125G-related samples. Moreover, Masson’s trichrome was also performed on the paraffin histological sections to better localize the fluorescent section within samples.

Additionally, Fil050G and Fil125G-related samples, implanted for 4 weeks and MMA-embedded from a previous study [40], were also used to quantify the number of blood vessels present in our constructs. Briefly, after slicing of the samples, sections were observed under a microscope with brightfield light to identify structures with lumens, and with fluorescent light to identify erythrocytes by their autofluorescence. The mean of both counts (Brightfield and Fluorescence) is provided.

### 4.9. Statistical Analysis

Results are expressed as mean ± SD. Before any statistical testing, the Shapiro–Wilk test was performed on all datasets to assess normal distribution. When the data did not satisfy the normality test, they were analyzed with the non-parametric Kruskal–Wallis test for multiple comparisons and Dunn’s post-hoc test or with Mann–Whitney test for single comparison. Datasets that passed the normality test were analyzed with 1-way ANOVA with Bonferroni’s or Dunnet’s post-test for multiple comparisons or with *t*-test for single comparison. Results were considered to be statistically significant at *p* values < 0.05 (* *p* < 0.05, ** *p* < 0.01, *** *p* < 0.001, and **** *p* < 0.0001). The data were processed with GraphPad Prism 5 Software (GraphPad Software Inc., San Diego, CA, USA).

## 5. Conclusions

Small changes in the microarchitecture of scaffolds by otherwise identical material, macroporosity, microporosity, directionality, and transparency, can have a profound effect on osteoconductivity and bony healing. Here we showed that the decrease in filament dimension and distance from 1.25 mm to 0.50 enhances angiogenesis, cell infiltration, and cell differentiation towards the osteogenic lineage at an early stage of bone healing. In later stages, this translates into high osteoconductivity and enhanced bony healing of the defect and improves the clinical outcome in orthopedic, craniofacial, and dental bone regeneration procedures.

## Figures and Tables

**Figure 1 ijms-24-06000-f001:**
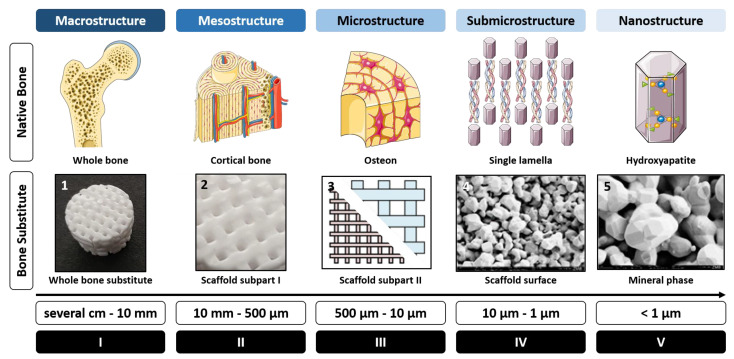
Five levels of hierarchical structure in native bone and bone substitute. (**I**) Macrostructure level (several cm to 10 mm), characterizing the whole bone tissue on one side or the whole bone substitute on the other side. (**II**) Mesostructure level (10 mm–0.5 mm), characterizing the cortical bone level on one side or scaffold subpart I (e.g., structure) on the other part. (**III**) Microstructure level (500 µm–10 µm), characterizing a single osteon on one side or scaffold subpart II (e.g., filament thickness) on the other side. (**IV**) Sub-microstructural level (10 µm–1 µm), characterizing the single lamella on one side and the scaffold surface (e.g., porosity) on the other side. (**V**) Nanostructure level (<1 µm), characterizing the hydroxyapatite structure on one side and the mineral phase (e.g., tricalcium phosphate) on the other side.

**Figure 2 ijms-24-06000-f002:**
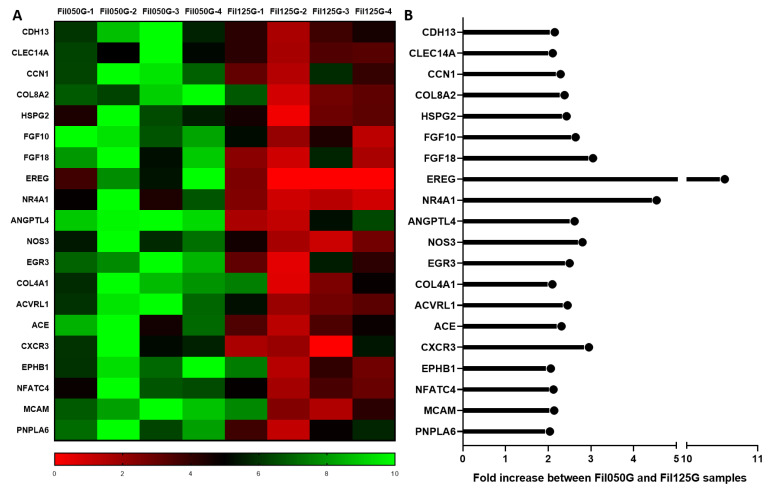
(**A**) Normalized expression of genes linked to angiogenesis in Fil050G and Fil125G samples (*n* = 4, each). (**B**) Fold increase of genes linked to angiogenesis in Fil050G and Fil125G samples.

**Figure 3 ijms-24-06000-f003:**
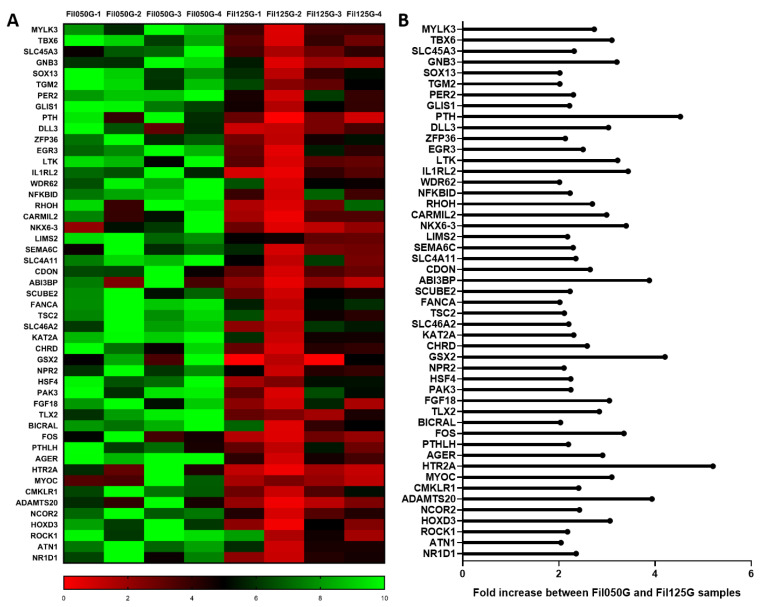
(**A**) Normalized expression of genes linked to regulation of cell differentiation in Fil050G and Fil125G samples (*n* = 4; each). (**B**) Fold increase of genes linked to regulation of cell differentiation in Fil050G and Fil125G samples.

**Figure 4 ijms-24-06000-f004:**
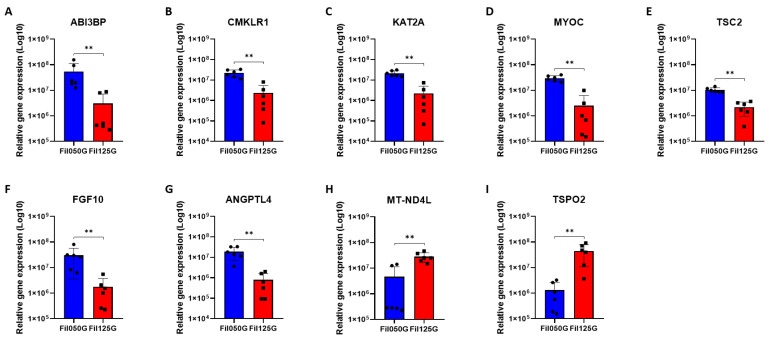
Relative expression of genes overrepresented with GO enrichment analysis in (**A**–**G**) Fil050G- and (**H**,**I**) Fil125G-related samples after 10 days of implantation; (**A**) ABI Family Member 3 Binding Protein, (**B**) Chemerin Chemokine-Like Receptor 1, (**C**) Lysine Acetyltransferase 2A, (**D**) Myocilin, (**E**) TSC Complex Subunit 2, (**F**) Fibroblast Growth Factor 10, (**G**) Angiopoietin-Like 4, (**H**) Mitochondrially Encoded NADH:Ubiquinone Oxidoreductase Core Subunit 4L, and (**I**) Translocator Protein 2. *p*-values are indicated as follow; ** *p* < 0.01.

**Figure 5 ijms-24-06000-f005:**
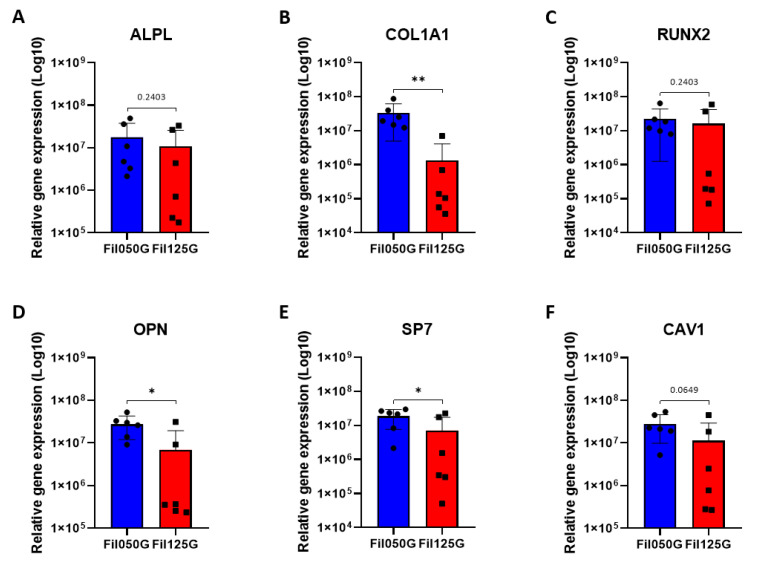
Relative expression of genes related to the osteogenic pathway in Fil050G- and Fil125G-related samples after 10 days of implantation; (**A**) Alkaline Phosphatase, (**B**) Collagen Type I Alpha 1 Chain, (**C**) RUNX Family Transcription Factor 2, (**D**) Osteopontin, (**E**) Osterix, and (**F**) Caveolin-1. *p*-values are indicated as follow; * *p* < 0.05, ** *p* < 0.01.

**Figure 6 ijms-24-06000-f006:**
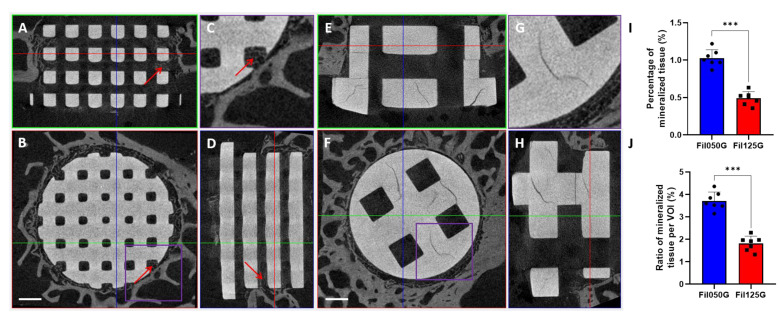
Illustration (**A**–**H**) and quantification (**I**,**J**) of µCT scanning for Fil050G- and Fil125G-related samples after 10 days of implantation. (**A**–**D**) Illustration of µCT scanning of Fil050G-related samples with side view (**A**,**D**), upper view (**B**), and magnification from the upper view (**C**). Illustration/representation of µCT scanning of Fil125G-related samples with side view (**E**,**H**), upper view (**F**), and a magnification from the upper view (**H**). Quantification of the percentage of mineralized tissue (**I**) and the ratio of mineralized tissue per volume of interest (**J**). Red arrows indicate mineralized tissue. Scale bar for (**A**–**H**) = 1 mm. *p*-values are indicated as follow; *** *p* < 0.001.

**Figure 7 ijms-24-06000-f007:**
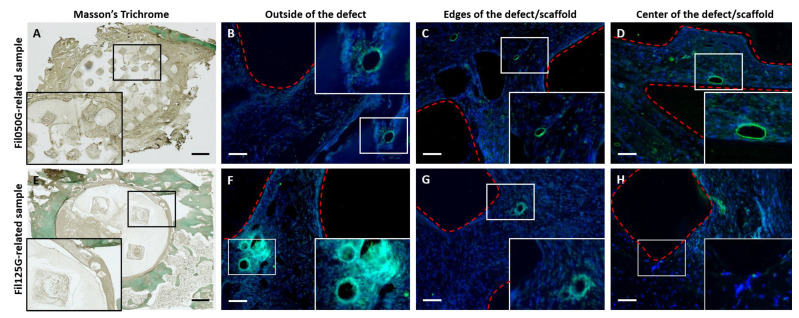
Histological staining with Masson’s Trichrome (**A**,**E**) and Laminin immunostaining (**B**–**D**,**F**–**H**) on Fil050G- and Fil125G-related samples after 10 days of implantation; (**A**) Histological staining of Fil050G-related samples with Masson’s Trichrome. (**B**,**F**) Laminin immunostaining outside of the defect/scaffold. (**C**,**G**) Laminin immunostaining at the edges of the defect/scaffold. (**C**,**H**) Laminin immunostaining at the center of the defect/scaffold. (**E**) Histological staining of Fil125G-related samples with Masson’s Trichrome. Laminin staining is shown as green (Alexa488) and DAPI in blue. Red dashed lines are delimiting the scaffold. Scale bar for (**A**,**E**) = 1 mm. Scale bar for (**B**–**D**,**F**–**H**) = 100 µm.

**Figure 8 ijms-24-06000-f008:**
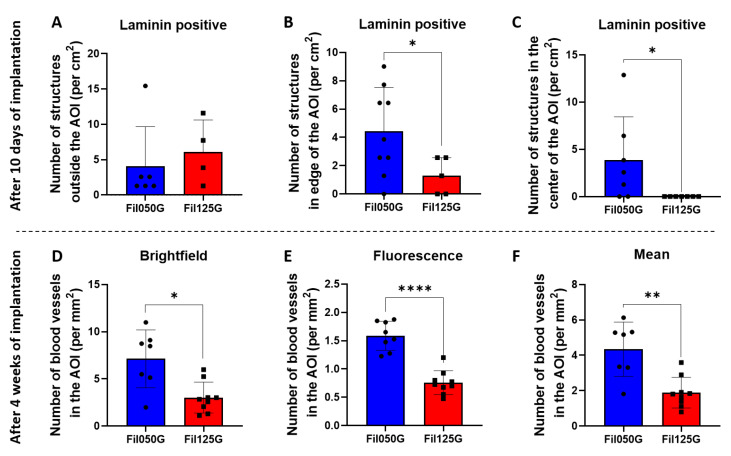
Laminin-positive structure (**A**–**C**) and blood vessel (**D**–**F**) quantification after 10 days and 4 weeks and implantation, respectively. (**A**) Number of Laminin-positive structures outside of the area of interest. (**B**) Number of Laminin-positive structures in the edge of the area of interest (**C**) Number of Laminin-positive structures in the center of the area of interest. (**D**) Number of blood vessels counted under brightfield light on histological section. (**E**) Number of blood vessels counted under fluorescent light on histological section. (**F**) Mean of counts under brightfield and fluorescent light on histological section. *p*-values are indicated as follow; * *p* < 0.05, ** *p* < 0.01, and **** *p* < 0.0001.

**Figure 9 ijms-24-06000-f009:**
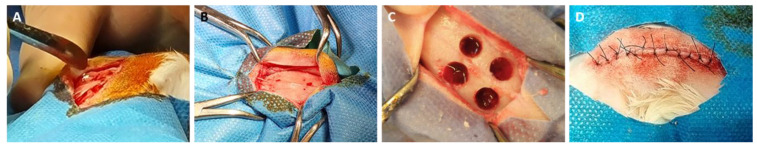
Step-by-step process of the surgical procedure. (**A**) The skin on the top of the cranium was disinfected and an incision was made from the nasal bone to the mid-sagittal crest. (**B**) The soft tissue was deflected and fixed, and the periosteum was removed. (**C**) Four defects were marked in the rabbit’s calvaria with a 6.00 mm trephine bur. To preserve the dura, they were then completed with rose burrs of 5.00 mm in diameter, followed by a burr with a 1.00 mm diameter. (**D**) Scaffolds were placed in the created bone defects and the wound was closed with sutures.

**Table 1 ijms-24-06000-t001:** Selected “biological process” terms significantly overrepresented in Fil050G and Fil125G after 10 days of implantation in the *Homo sapiens* (*Hs*), *Mus musculus* (*Mm*), and *Rattus norvegicus* (*Rn*) databases.

Biological Pathways Overrepresented in 3 Out of 3 Databases	Biological Pathways Overrepresented in 2 Out of 3 Databases	Biological Pathways Overrepresented in 1 Out of 3 Databases
Adaptive immune response	Cell motility (*Rn*, *Mm*)	Cell surface receptor signaling pathway (*Rn*)
Positive regulation of cell adhesion	Cellular process (*Rn*, *Mm*)	Dendrite morphogenesis (*Rn*)
Regulation of cell adhesion	Negative regulation of cell communication (*Rn*, *Mm*)	Detection of chemical stimulus (*Rn*)
Cell adhesion	Negative regulation of signaling (*Rn*, *Mm*)	Detection of chemical stimulus involved in sensory perception (*Rn*)
Regulation of cell migration	Nervous system development (*Rn*, *Mm*)	Detection of chemical stimulus involved in sensory perception of smell (*Rn*)
Cell–matrix adhesion	Positive regulation of cell communication (*Rn*, *Mm*)	Detection of stimulus (*Rn*)
Cell–substrate adhesion	Positive regulation of signal transduction (*Rn*, *Mm*)	Detection of stimulus involved in sensory perception (*Rn*)
Anatomical structure development	Regulation of intracellular signal transduction (*Rn*, *Mm*)	Intracellular receptor signaling pathway (*Rn*)
Anatomical structure formation involved in morphogenesis	Regulation of response to stimulus (*Rn*, *Mm*)	Negative regulation of response to stimulus (*Rn*)
Anatomical structure morphogenesis	Sensory perception of chemical stimulus (*Rn*, *Mm*)	Positive regulation of cellular metabolic process (*Rn*)
Animal organ development	Sensory perception of smell (*Rn*, *Mm*)	Positive regulation of macromolecule metabolic process (*Rn*)
Developmental process	System development (*Rn*, *Mm*)	Positive regulation of metabolic process (*Rn*)
Enzyme-linked receptor protein signaling pathway	Cell motility (*Rn*, *Mm*)	Positive regulation of nucleobase-containing compound metabolic process (*Rn*)
Multicellular organismal process	Cellular process (*Rn*, *Mm*)	Positive regulation of response to stimulus (*Rn*)
Muscle system process	Negative regulation of cell communication (*Rn*, *Mm*)	Regulation of cell motility (*Rn*)
Regulation of cell communication	Negative regulation of signaling (*Rn*, *Mm*)	Regulation of cell population proliferation (*Rn*)
Regulation of multicellular organismal process	Nervous system development (*Rn*, *Mm*)	Regulation of IP-10 production (*Rn*)
Regulation of signal transduction	Positive regulation of cell communication (*Rn*, *Mm*)	Cell projection organization (*Mm*)
Regulation of signaling	Positive regulation of signal transduction (*Rn*, *Mm*)	Negative regulation of response to stimulus (*Mm*)
	Regulation of intracellular signal transduction (*Rn*, *Mm*)	Cellular response to endogenous stimulus (*Mm*)

**Table 2 ijms-24-06000-t002:** Selected “biological process” terms significantly overrepresented solely in Fil050G at 10 days of implantation in the *Homo sapiens* (*Hs*), *Mus musculus* (*Mm*), and *Rattus norvegicus* (*Rn*) databases.

Biological Pathways Overrepresented in 3 Out of 3 Databases	Biological Pathways Overrepresented in 2 Out of 3 Databases	Biological Pathways Overrepresented in 1 Out of 3 Databases
Circulatory system development	Angiogenesis (*Rn*, *Mm*)	Blood circulation (*Rn*)
Phospholipid catabolic process	Cell development (*Rn*, *Mm*)	Blood vessel development (*Rn*)
Skeletal muscle cell differentiation	Cell differentiation (*Rn*, *Mm*)	Detection of stimulus involved in sensory perception (*Rn*)
Tissue development	Embryonic morphogenesis (*Rn*, *Mm*)	Embryo development (*Rn*)
	Multicellular organism development (*Rn*, *Mm*)	Formation of primary germ layer (*Rn*)
	Positive regulation of biological process (*Rn*, *Mm*)	Heart development (*Rn*)
	Positive regulation of cellular process (*Rn*, *Mm*)	Mesoderm formation (*Rn*)
	Regulation of cell differentiation (*Rn*, *Mm*)	Mesoderm morphogenesis (*Rn*)
	Regulation of developmental process (*Rn*, *Mm*)	Muscle tissue development (*Rn*)
	Regulation of MAPK cascade (*Rn*, *Mm*)	Positive regulation of signaling (*Rn*)
	Tube morphogenesis (*Rn*, *Mm*)	Regulation of biological quality (*Rn*)
		Regulation of cell migration (*Rn*)
		Regulation of ERK1 and ERK2 cascade (*Rn*)
		Skeletal muscle tissue development (*Rn*)
		Tissue morphogenesis (*Rn*)
		Animal organ morphogenesis (*Mm*)
		Cell adhesion (*Mm*)

**Table 3 ijms-24-06000-t003:** Selected GO “molecular function” terms significantly overrepresented solely in Fil050G after 10 days of implantation in the *Homo sapiens* (*Hs*), *Mus musculus* (*Mm*), and *Rattus norvegicus* (*Rn*) databases.

Molecular Functions Overrepresented in 3 Out of 3 Databases	Molecular Functions Overrepresented in 2 Out of 3 Databases	Molecular Functions Overrepresented in 1 Out of 3 Databases
Phospholipase activity	Calcium-dependent phospholipase A2 activity (*Rn*, *Mm*)	Actin binding (*Rn*)
Lipase activity	Phospholipase A2 activity (*Rn*, *Mm*)	Ion binding (*Mm*)
External encapsulating structure	Carboxylic ester hydrolase activity (*Rn*, *Mm*)	Odorant binding (*Mm*)
	Nuclear glucocorticoid receptor binding (*Rn*, *Mm*)	
	Protein binding (*Rn*, *Mm*)	
	Binding (*Rn*, *Mm*)	
	Metal ion binding (*Rn*, *Mm*)	
	Cation binding (*Rn*, *Mm*)	
	Olfactory receptor activity (*Rn*, *Mm*)	

**Table 4 ijms-24-06000-t004:** Selected GO “cellular component” terms significantly overrepresented solely in Fil050G after 10 days of implantation in the *Homo sapiens* (*Hs*), *Mus Musculus* (*Mm*), and *Rattus Norvegicus* (*Rn*) databases.

Cellular Components Overrepresented in 3 Out of 3 Databases	Cellular Components Overrepresented in 2 Out of 3 Databases	Cellular Components Overrepresented in 1 Out of 3 Databases
Collagen-containing extracellular matrix	Plasma membrane bounded cell projection (*Rn*, *Mm*)	Sarcolemma (*Rn*)
Extracellular matrix	Cell projection (*Rn*, *Mm*)	Extracellular region (*Rn*)
External encapsulating structure	Cellular anatomical entity (*Rn*, *Mm*)	Ribonucleoprotein complex (*Rn*)
		Caveola (*Rn*)
		Cell periphery (*Mm*)

**Table 5 ijms-24-06000-t005:** Genes upregulated in the angiogenesis and regulation of cell differentiation pathways associated with ossification and bone development in “biological process” terms with their fold increase.

Fold Change	Symbol	Full Name
3.047949166	FGF18	Fibroblast growth factor 18
3.097075048	MYOC	Myocilin
2.23006642	SCUBE2	Signal peptide, CUB and EGF-like domain-containing protein 2
2.103939633	NPR2	Atrial natriuretic peptide receptor 2
2.017002613	TGM2	Protein-glutamine gamma-glutamyltransferase 2
2.288859577	CCN1	CCN family member 1
2.194750496	PTHLH	Parathyroid hormone-related protein
2.219050295	GLIS1	Zinc finger protein GLIS1
4.521904971	PTH	Parathyroid hormone
2.429593561	HSPG2	Basement membrane-specific heparan sulfate proteoglycan core protein
2.584840247	CHRD	Chordin
2.802297373	NOS3	Nitric oxide synthase, endothelial

**Table 6 ijms-24-06000-t006:** List of primers used in this study, with their short name, full name, Ensembl number, NCBI Entrez Gene number, and BioRad unique assay ID.

Short Name	Full Name	Ensembl Number; NCBI Entrez Gene Number	BioRad Unique Assay ID
GAPDH	Glyceraldehyde-3-Phosphate Dehydrogenase	ENSOCUG00000025023; 100009074	qOcuCED0019227
18S	Ribosomal Protein S18	ENSOCUG00000025023; 100009074	qOcuCED0018100
ABI3BP	ABI Family Member 3 Binding Protein	ENSOCUG00000014699; 100349921	qOcuCID0005404
CMKLR1	Chemerin Chemokine-Like Receptor 1	ENSOCUG00000005336; 100358278	qOcuCED0007914
KAT2A	Lysine Acetyltransferase 2A	ENSOCUG00000012259; 100355497	qOcuCED0011401
MYOC	Myocilin	ENSOCUG00000013285; 100347774	qOcuCED0012306
TSC2	TSC Complex Subunit 2	ENSOCUG00000015957; 100008897	qOcuCID0001468
FGF10	Fibroblast Growth Factor 10	ENSOCUG00000012834; 100351640	qOcuCED0010180
ANGPTL4	Angiopoietin-Like 4	ENSOCUG00000008499; 100342802	qOcuCED0008856
MT-ND4L	Mitochondrially Encoded NADH:Ubiquinone Oxidoreductase Core Subunit 4L	ENSOCUG00000016603; 100339191	qOcuCED0019933
TSPO2	Translocator Protein 2	ENSOCUG00000029107; 808222	qOcuCED0019792
ALPL	Alkaline Phosphatase, Biomineralization Associated	ENSOCUG00000027644; 100347991	qOcuCED0012587
COL1A1	Collagen Type I Alpha 1 Chain	ENSOCUG00000004447; 100341109	qOcuCED0015663
RUNX2	RUNX Family Transcription Factor 2	ENSOCUG00000012881; 100347598	qOcuCID0006259
OPN	Osteopontin, Secreted Phosphoprotein 1	ENSOCUG00000011739; 100008943	qOcuCED0010864
SP7	Osterix, Sp7 Transcription Factor	ENSOCUG00000022536; 100008982	qOcuCED0015349
CAV1	Caveolin 1	ENSOCUG00000003811; 100356871	qOcuCID0002632

## Data Availability

Data will not be available due to privacy restrictions.

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
