# Peer review of "Influence of Scaffold Microarchitecture on Angiogenesis and Regulation of Cell Differentiation during the Early Phase of Bone Healing: A Transcriptomics and Histological Analysis"

_ijms, 2023, doi:10.3390/ijms24066000_

Round 1

Reviewer 1 Report

The article describes the underlying mechanisms of bone healing via the prism of scaffold microarchitecture and its role on the early stages of regeneration. The article provides novel data and has significant scientific soundness and interest. However, the following improvements are recommended before considering the article for publication:

1. The authors are recommended to support the Materials & Methods section with a couple of intraoperative pictures of the wound and implantation site for better understanding of surgical procedure

2.  A SEM-images of scaffolds could be added as a provement of required and expected porosity.

Author Response

Comments to the Author

Reviewer 1:

    The article describes the underlying mechanisms of bone healing via the prism of scaffold microarchitecture and its role on the early stages of regeneration. The article provides novel data and has significant scientific soundness and interest. However, the following improvements are recommended before considering the article for publication:

Answer: We appreciate the positive feedback on our manuscript. We thank the reviewer for a constructive and relevant input. We have carefully considered the comments and addressed them as indicated below.

  1. The authors are recommended to support the Materials & Methods section with a couple of intraoperative pictures of the wound and implantation site for better understanding of surgical procedure.

Answer: Thank you for pointing this out. We agree with the reviewer's comment on implementing the Materials and Method section with pictures and a better description of the intraoperative procedure as well as the wound and implantation site. A new Figure (i.e. Figure 9) was added on Page 16 to illustrate and document the procedure. The Figure caption was also added on Page 16 and Page 17 and will be as follow:

Figure 9: Step-by-step process of the surgical procedure. (A) The skin on the top of the cranium was disinfected and an incision was made from the nasal bone to the mid-sagittal crest. (B) The soft tissue was deflected and fixed, and the periosteum was removed. (C) Four defects were marked in the rabbit's calvaria with a 6.00 mm trephine bur. To preserve the dura, they were then completed with rose burrs of 5.00 mm in diameter, followed by a burr with a 1.00 mm diameter. (D) Scaffolds were placed in the created bone defects and the wound was closed with sutures.

  1. A SEM-images of scaffolds could be added as a provement of required and expected porosity.

Answer: Thank you for mentioning this point. We already illustrated and documented all characteristics from those two scaffolds in another recent publication from our lab by Guerrero et al., in 2022 (PMID: 36844242, reference n°40 in our study). Based on the design, the overall porosity is 50% and the same for both constructs. The summarized data from the cited publication are the following for both scaffolds;

Microarchitecture

Filaments in direction of bone growth (%)

Rod dimension and rod distance (mm)

Macroporosity (%)

Transparency (%)

Fil050G

50

0.50

50

25

Fil125G

50

1.25

50

25

We pointed the similarity of the 2 constructs concerning porosity, microporosity, transparency out in the introduction.

In a recent study, those two scaffolds were established with marked differences in their osteoconductivity resulting in high (Fil050G) or low (Fil125G) osteoconductive scaffolds, although porosity, microporosity, transparency and degree of directionality of the filaments was identical [40].

Comments to the Author

Reviewer 1:

    The article describes the underlying mechanisms of bone healing via the prism of scaffold microarchitecture and its role on the early stages of regeneration. The article provides novel data and has significant scientific soundness and interest. However, the following improvements are recommended before considering the article for publication:

Answer: We appreciate the positive feedback on our manuscript. We thank the reviewer for a constructive and relevant input. We have carefully considered the comments and addressed them as indicated below.

  1. The authors are recommended to support the Materials & Methods section with a couple of intraoperative pictures of the wound and implantation site for better understanding of surgical procedure.

Answer: Thank you for pointing this out. We agree with the reviewer's comment on implementing the Materials and Method section with pictures and a better description of the intraoperative procedure as well as the wound and implantation site. A new Figure (i.e. Figure 9) was added on Page 16 to illustrate and document the procedure. The Figure caption was also added on Page 16 and Page 17 and will be as follow:

Figure 9: Step-by-step process of the surgical procedure. (A) The skin on the top of the cranium was disinfected and an incision was made from the nasal bone to the mid-sagittal crest. (B) The soft tissue was deflected and fixed, and the periosteum was removed. (C) Four defects were marked in the rabbit's calvaria with a 6.00 mm trephine bur. To preserve the dura, they were then completed with rose burrs of 5.00 mm in diameter, followed by a burr with a 1.00 mm diameter. (D) Scaffolds were placed in the created bone defects and the wound was closed with sutures.

  1. A SEM-images of scaffolds could be added as a provement of required and expected porosity.

Answer: Thank you for mentioning this point. We already illustrated and documented all characteristics from those two scaffolds in another recent publication from our lab by Guerrero et al., in 2022 (PMID: 36844242, reference n°40 in our study). Based on the design, the overall porosity is 50% and the same for both constructs. The summarized data from the cited publication are the following for both scaffolds;

Microarchitecture

Filaments in direction of bone growth (%)

Rod dimension and rod distance (mm)

Macroporosity (%)

Transparency (%)

Fil050G

50

0.50

50

25

Fil125G

50

1.25

50

25

We pointed the similarity of the 2 constructs concerning porosity, microporosity, transparency out in the introduction.

In a recent study, those two scaffolds were established with marked differences in their osteoconductivity resulting in high (Fil050G) or low (Fil125G) osteoconductive scaffolds, although porosity, microporosity, transparency and degree of directionality of the filaments was identical [40].

Reviewer 2 Report

The writing of the article is very neat, each part is coherent, clearly presented and very well argued.

The numerous and relevant results clearly show the desired objectives.

Just one question: did you notice a significant difference in the pore volume between the 2 Fil050G and Fil125G samples?

 I congratulate the authors for the excellent scientific quality and relevance of this work.

Author Response

Reviewer 2:

    The writing of the article is very neat, each part is coherent, clearly presented and very well argued. The numerous and relevant results clearly show the desired objectives. I congratulate the authors for the excellent scientific quality and relevance of this work.

Answer: We would like to thank the reviewer for a very positive evaluation of our study.

  1. Did you notice a significant difference in the pore volume between the 2 Fil050G and Fil125G samples?

Answer: Thanks for highlighting this point. As mentioned with the previous reviewer, we already documented all characteristics from those two scaffolds in another recent publication from our lab by Guerrero et al., in 2022 (PMID: 36844242, reference n°40 in our study). Based on the design overall porosity is 50% and the same for both constructs. The summarized data from the cited publication are the following for both scaffolds;

Microarchitecture

Filaments in direction of bone growth (%)

Rod dimension and rod distance (mm)

Macroporosity (%)

Transparency (%)

Fil050G

50

0.50

50

25

Fil125G

50

1.25

50

25

We pointed the similarity of the 2 constructs concerning porosity, microporosity, transparency out in the introduction.

In a recent study, those two scaffolds were established with marked differences in their osteoconductivity resulting in high (Fil050G) or low (Fil125G) osteoconductive scaffolds, although porosity, microporosity, transparency and degree of directionality of the filaments was identical [40].
